



# A global reference database in FAOSTAT of cropland nutrient budgets and nutrient use efficiency: nitrogen, phosphorus and potassium, 1961-2020.

Cameron I. Ludemann[1], Nathan Wanner[2], Pauline Chivenge[3], Achim Dobermann[4], Rasmus Einarsson[5], Patricio Grassini[6], Armelle Gruere[4], Kevin Jackson[7], Luis Lassaletta[8], Federico Maggi[9], Griffiths Obli-Laryea[2], Martin K. van Ittersum[1], Srishti Vishwakarma[7,10],Xin Zhang[7], Francesco N. Tubiello[2,*]

[1] Plant Production Systems Group, Wageningen University & Research, Wageningen, Netherlands
[2] Statistics Division, Food and Agriculture Organization, Rome, Italy
[3]African Plant Nutrition Institute, Benguérir, Morocco
[4] International Fertilizer Association, Paris, France
[5] Swedish University of Agricultural Sciences, Uppsala, Sweden
[6] University of Nebraska-Lincoln, Lincoln, USA
[7] University of Maryland Center for Environmental Science, Appalachian Laboratory, Maryland, USA
[8] CEIGRAM/Agricultural Production. ETSIAAB. Universidad Politécnica de Madrid, Madrid, Spain
[9] Environmental Engineering, School of Civil Engineering, The University of Sydney, 2006, NSW, Australia
[10]Currently located at the Joint Global Change Research Institute, Pacific Northwest National Laboratory, Maryland, USA.

*Corresponding author: francesco.tubiello@fao.org

**Abstract.** Nutrient budgets help to identify excess or insufficient use of fertilizers and other nutrient sources in agriculture. They allow calculation of indicators such as the nutrient balance (surplus or deficit) and nutrient use efficiency that help in monitoring of agricultural productivity and sustainability across the world. We present a global database of country-level budget estimates for nitrogen (N), phosphorus (P) and potassium (K) in cropland. The database, disseminated in FAOSTAT, 25 is meant to provide a global reference, synthesizing and continuously updating the state-of-the-art on this topic. The database covers 205 countries and territories, as well as regional and global aggregates, for the period 1961 to 2020. Results highlight the wide range in nutrient use and use efficiencies across geographic regions, nutrients, and time. For the year 2020, the data show regional average N surpluses that range from about 10 kg N ha$^{-1}$ year$^{-1}$ in Africa to more than 90 kg N ha$^{-1}$ year$^{-1}$ in Asia. Furthermore, they highlight P and K deficits in Africa in 2020 and K deficits for the Americas. This study introduces 30 improvements over previous work in relation to key nutrient coefficients affecting nutrient budgets and use efficiency estimates, especially for nutrient removal in crop products, manure nutrient content, atmospheric deposition and crop biological N fixation rates. We conclude by discussing future research directions, highlighting the need to align statistical definitions across research groups, as well as to further refine plant and livestock coefficients and expand estimates to all agricultural land, including nutrient flows in meadows and pastures.




## 1 Introduction

Nutrient budgets quantify nutrient flows in agriculture and are widely used to quantify the productivity and resource use efficiency of agricultural systems. The nutrient surplus, calculated as the difference between nutrient inputs and outputs, is an indicator of excess or insufficient use of nutrients from fertilizers and other sources in crop production. Nutrient surpluses
threaten environmental quality, particularly with regard to water and air quality, climate change, and biodiversity loss (Zhang et al., 2021; FAO, 2022a). On the other hand, nutrient deficits, or nutrient surpluses close to zero could indicate soil nutrient mining, potentially decreasing soil health over time. Imbalanced crop nutrition endangers the productivity and sustainability of agriculture. Comparable data on soil nutrient budgets and related indicators of nutrient use efficiency are therefore useful tools to assess and monitor agricultural performance and may support the 2030 Sustainable Development Goals indicators
(Zhang et al., 2021; FAO, 2022a; Quan et al., 2021; Tubiello et al., 2021).

Data presented in this work focus on *partial* nutrient budgets (referred herein to as nutrient budgets) and related nutrient use efficiencies on cropland. The term *partial* is here used to indicate that what is computed herein is in fact a partial nutrient budget in which specific nutrient losses such as gaseous emissions, leaching or runoff are not explicitly accounted for. In other words, such losses are embedded in the overall nutrient budget estimates, whereas a *net* nutrient budget would explicitly
include specific estimates of the different losses. Cropland is the sum of arable land and permanent crops, including areas left fallow or cultivated with temporary pastures within crop rotations, but excluding permanent meadows and pastures (FAO, 2022c). We see two main rationales for estimating nutrient budgets on cropland. First, cropland is typically where nutrient flows and related environmental impacts are highest, and cropland budgets and derived indicators such as the surplus are therefore more likely to capture potential pollution hotspots. Second, permanent meadows and pastures present some particular
method challenges, primarily due to lack of global data on productivity and biological N fixation.

The nutrient budget inputs in cropland considered in this work are the application of synthetic fertilizers (also referred to as "chemical fertilizers" or "mineral fertilizers"), manure from livestock, and the N inputs through biological N fixation and the atmospheric N deposition. The nutrient budget outputs are the nutrients removed via crop harvest. The nutrient surplus (also known as balance or, in the case of a negative surplus, deficit) is calculated as the difference between inputs and outputs.
Finally, nutrient use efficiency is computed as nutrient outputs as a percentage of nutrient inputs.

Data presented here build on previous work on estimating national to global scale nutrient budgets (or important components of nutrient budgets) over time (Einarsson et al., 2020; Oenema et al., 2003; FAO, 2021; Peoples et al., 2021; Ludemann et al., 2022a; Herridge et al., 2022; Kremer, 2013; Vishwakarma et al., 2022; IFA, 2022b; Zhang et al., 2021; Einarsson et al., 2021; Zou et al., 2022; Zhang et al., 2015; Conant et al., 2013; Lassaletta et al., 2014; Lassaletta et al., 2016; Mueller et al., 2012;
Bodirsky et al., 2012; Bouwman et al., 2013; Lu and Tian, 2017; Nishina et al., 2017). It adds additional refinements—such as new estimates of synthetic fertilizer inputs, the fraction of fertilizer applied to cropland, manure, N deposition, biological N fixation, and nutrients removed in harvested crops (see Methods). The new data are made available freely to users worldwide for N, P and K budgets, budget components and nutrient use efficiencies, covering 205 countries and territories for the period



1961-2020 (FAO, 2022d). The resulting dataset represents in our view the most complete dataset so far on the subject matter, serving as a reference for additional refinements by the scientific community.

## 2 Methods

The Cropland Nutrient Budget (CNB) was developed for N, P and K data at country level, for all areas of cropland as a FAO land use category (FAO, 2022a, c). The nutrient surplus or deficit for country i, nutrient j and year y, was computed as the sum of inputs: synthetic fertilizers (SF) multiplied by the fraction of fertilizer applied to cropland (CF), manure applied to cropland soils (MAS), atmospheric deposition (AD; only for N), and biological N fixation (BF; only for N) minus crop removal (CR), which represents the outputs in the CNB (Equation 1, Figure 1). Positive values are referred to as a surplus, while negative values are referred to as a deficit.

$$surplus/deficit_{i,j,y} = SF_{i,j} \times CF_{i,j,y} + MAS_{i,j,y} + AD_{i,j,y} + BF_{i,j,y} - CR_{i,j,y} \qquad (1)$$

Data are computed both as total nutrients and on a per area of cropland per year basis. Collection and analysis of each of these CNB components are described in more detail in the following sections.

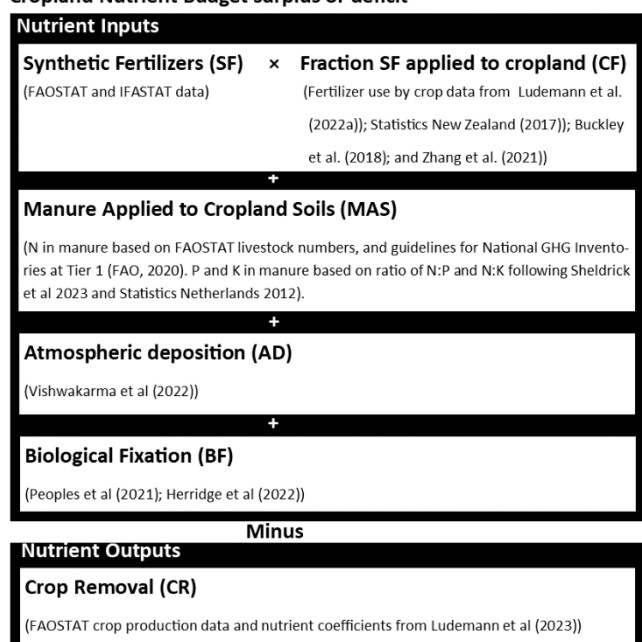

**Figure 1: Cropland nutrient budget components.**



### 2.1 Cropland area

Cropland area defines the scope for the estimations made in this work. Cropland herein is defined as the land use category, defined by FAO for collection of country data (https://www.fao.org/statistics/data-collection/en/) as the 'land used for cultivation of temporary and permanent crops in rotation with fallow, meadows and pastures within cycles of up to five years.' It is important to underscore that the land use term 'cropland' in general encompasses more area than the corresponding term used in remote sensing and bio-physical modelling, which largely refers only to land areas planted or harvested with annual

crops (Tubiello et al., 2023). Identifying flows on cropland as a land use category allows for clear operational definitions of what is in scope with regards to CNB data at country level in line with FAO reporting. At the same time, it generates significant uncertainty in the associated quantities, as discussed below.

### 2.2 Cropland nutrient budget components

Information for SF inputs were sourced from data on agricultural use from both FAOSTAT (FAO, 2022b) and IFASTAT (IFA,

2022b), taking the mean value of the two data sources when both were available. The individual datasets have been shown to be rather equivalent (FAO, 2022a), so that in a reference database the choice was on a consolidated dataset from both sources. All SF values were converted to elemental quantities of nutrients based on a mass proportion composition conversion of 0.436kg elemental P per kg $P_2O_5$ and 0.83kg elemental K per kg of $K_2O$. Where necessary, all other inputs and outputs were converted to quantities of elemental nutrients using these conversion factors.

Importantly, both FAO and IFA data refer to fertilizer use in agriculture generally, while actual amounts used specifically on cropland are not systematically estimated. The fraction of fertilizer applied to cropland (CF) was therefore needed to determine inputs for the CNB developed in this work. For the majority of countries, due to lack of specific information, default cropland fraction estimates of 100% were used for N, P, and K, thereby assuming all fertilizers were applied on cropland area. At the same time, we were able to identify 21 countries for which reasonable evidence is available to support specific values of CF

for N (Table 1). CF for major crops by country were first estimated for N considering estimates derived from 4 sources (Ludemann 2022a; FAO 2022; Einarsson et al., 2021; Zhang et al., 2021). The 21 countries  with new CF estimates were selected based on relatively stringent criteria; namely, if a given country: (1) had reported CF estimates for N from IFA and/or FAO, (2) that selected CF estimates for N use were significantly lower than 100%, and that (3) CF estimates were in general good agreement across these various sources. In addition, for two countries, Ireland and New Zealand, we used the CF values

communicated by the country directly to FAO as its part of statistical data collection. Conversely, default CF values were used for countries where: (1) there was lack of sufficient data, (2) reported estimates were close to 100% (e.g., >90%), or if (3) there existed disagreement in reported values by our available sources. For countries with recommended updates, CF for P was based on reported values by  Zou et al. (2022). The CF for K were calculated as averages of the N and P coefficients. Further clarification on the derivation and screening of CF estimates are included in Supplementary Material 1.




**Table 1: Percentages (%) of total nitrogen (N), phosphorus (P) and potassium (K) fertilizers used in agriculture\* applied to cropland, for selected countries and years.**

| Country | N | P | K |
|---|---|---|---|
| Australia | 90 | 70 | 80 |
| Austria | 90 | 90 | 90 |
| Brazil | 90 | 100 | 95 |
| Canada | 90 | 100 | 95 |
| Chile | 80 | 70 | 75 |
| Finland | 70 | 100 | 85 |
| France | 90 | 90 | 90 |
| Germany | 80 | 90 | 85 |
| Ireland | 20 | 30 | 25 |
| Japan | 80 | 100 | 90 |
| Morocco | 90 | 100 | 95 |
| Netherlands | 50 | 90 | 70 |
| New Zealand | 10 | 10 | 10 |
| Poland | 80 | 90 | 85 |
| Slovenia | 60 | 70 | 65 |
| South Africa | 90 | 90 | 90 |
| Switzerland | 70 | 70 | 70 |
| United Kingdom | 80 | 70 | 75 |
| United States | 80 | 100 | 90 |
| Uruguay | 90 | 90 | 90 |
| Luxembourg | 40 | 70 | 55 |
| Other countries | 100 | 100 | 100 |

\*These percentages were also used to apportion nutrients

from manure from livestock in agriculture to cropland

Organic N inputs were limited to manure applied to cropland soils (MAS), with N amounts estimated by FAO using the IPCC
Guidelines for National Greenhouse Gas Emission Inventories at Tier 1, whereby default live animal weights were multiplied
by N excretion coefficients (e.g., FAO (2020)). The associated P and K quantities were subsequently estimated using published
P:N and K:N conversion ratios (Statistics Netherlands, 2012; Sheldrick et al., 2003)(Table 2). The N, P and K nutrients from
manure from livestock were apportioned to cropland based on the same CF values shown in Table 1.





**Table 2: Phosphorus (P) and potassium (K) to N ratios in livestock manure**

| FAO code | Item | P ratio | K ratio |
|---|---|---|---|
| 960 | Cattle, dairy | 0.14 | 1.11 |
| 961 | Cattle, non-dairy | 0.19 | 0.95 |
| 976 | Sheep | 0.16 | 0.96 |
| 1016 | Goats | 0.17 | 0.88 |
| 1049 | Swine, market | 0.25 | 0.55 |
| 1051 | Swine, breeding | 0.28 | 0.45 |
| 1052 | Chickens, layers | 0.27 | 0.37 |
| 1053 | Chickens, broilers | 0.22 | 0.34 |
| 1068 | Ducks | 0.18 | 0.32 |
| 1079 | Turkeys | 0.23 | 0.33 |
| 1096 | Horses | 0.18 | 0.80 |
| 1759 | Mules and Asses | 0.18 | 0.80 |
| 1760 | Camels and Llamas | 0.18 | 0.80 |
| 946 | Buffaloes | 0.16 | 1.17 |

Atmospheric deposition (AD) refers to N inputs from the atmosphere as dry and wet N deposition considering both the reduced and oxidised forms, which was derived from a review of existing methods and related data sources for national scale data as

described by Vishwakarma et al. (2022). Out of the four datasets for AD, the product comprising of LUH2 (Hurtt et al., 2020) and Wang et al (Wang et al., 2019; Shang et al., 2019; Wang et al., 2017) data were used in the CNB. The flows of P and K through atmospheric deposition are generally negligible (Einarsson et al., 2020) so were not included in the CNB.

Biological fixation (BF) of N by grain legume crops was estimated using a yield-dependent and regionally-specific model presented by Peoples et al. (2021) and Herridge et al. (2022). This model was based on statistical regressions for eight

categories of grain legumes (chickpea, common bean, faba bean, groundnut, lupin, pigeon pea, soybean, and other). For soybeans, the model further distinguishes Brazil, Europe, and the rest of the world. The model assumes a non-linear dependence of BF rate on crop yield, and therefore, in contrast to earlier publications, does not lead to fixed ratios between harvest area and BF. Further details of these models are included in Supplementary Material 2. Forage legumes were not accounted for due to lack of production data (see Section 2.3.1 below). For non-legume crops BF was estimated using fixed

global per-hectare coefficients of 25 kg N ha$^{-1}$ year$^{-1}$ for rice and sugar cane (see Supplementary Material 2 for detailed explanation). With the exception of rice and sugar cane, N fixation from free-living N fixing bacteria in other crops were not included in the CNB. Source code (in R and Python) as well as detailed output for the BF estimates are freely available (Einarsson, 2023b, a).



Crop removal (CR) rates were calculated through crop nutrient removal coefficients multiplied by crop production statistics.
The crop production data were taken from FAOSTAT (2022b). Crop nutrient removal coefficients from Supplemental
Material 3 were used to estimate total crop nutrient removal, which were derived from a meta-analysis described by Ludemann
et al. (2023a). The source code for standardizing and analysing the CR data in R was published separately by Ludemann
(2022).

The same aforementioned coefficients for all the CNB components were applied to each year across the full time series.

**2.3 Data limitations and uncertainty**

**2.3.1 Scope**

The nutrient budgets presented here refer to the FAO cropland area (as defined in Section 2.1), while acknowledging that there
is substantial uncertainty in its measurement and a variety of definitions across various relevant land cover products (Tubiello
et al., 2023). The world's cropland area used in the present study was taken from FAOSTAT and was 1.562 billion ha for
2020. This compares with the 1.215-2.002 billion ha range and a ~25% relative uncertainty in cropland area recently estimated
by Tubiello et al. (2023) (Table 3 and Supplementary Material 4). In addition, the CNB excludes crops with no production
data in FAOSTAT (2022b). These include forage crops such as alfalfa, clover and grass-clover mixtures. Exclusion of these
crops likely leads to substantial underestimation of cropland nutrient removal, and in some cases cropland biological N fixation,
in countries where forage legumes are major components of cropland, such as Australia, Argentina, several European countries
(Einarsson et al., 2021), New Zealand and the United States of America. Another cause of uncertainty in the CNB arises from
how the parameters were estimated, as is described in the next section.

**Table 3: Estimates of relative uncertainty (expressed as the coefficient of variation-CV%) in key items and affected components of
the Cropland Nutrient Budget (CNB) using 2020 data. Details of each contributing item and component are included in
Supplementary Material 4.**

| Item | Components of CNB item effects* | Relative uncertainty (%)** |
|---|---|---|
| Cropland area | All | 25% |
| Crop production | CR, BF | 7% |
| Livestock numbers | MAS | 10% |
| Livestock manure nutrient coefficients | MAS | 50% |
| Synthetic fertilizer (SF) use | SF | 25% |
| Fraction of SF applied to cropland | SF | 10% |
| Atmospheric deposition of N | AD | 70% |
| Biological N fixation coefficients | BF | 60% |
| Crop removal coefficients | CR | 20% |



*Components of Cropland Nutrient Budget (CNB) include: synthetic fertilizers (SF), fraction of SF applied to cropland (CF), manure applied to soils (MAS), atmospheric deposition (AD), biological fixation (BF), crop removal (CR).**Uncertainty was expressed as the coefficient of variation to 2 significant figures.

### 2.3.2 Uncertainty

Nutrient budgets tend to have large uncertainties (Lesschen et al., 2007; Pathak et al., 2010). However, in general, there appears to be more certainty in the direction (e.g. is it a negative or positive budget) than in the magnitude of nutrient budgets. For example, where multiple studies estimated the N, P and K budgets for Burkina Faso, there was good concordance (90%

showing same direction) in whether there was a deficit or surplus (Lesschen et al., 2007). However, the coefficient of variation of these estimates of nutrient budgets in Burkina Faso made by the various researchers was 27 % for N, 167% for P and 115% for K (Lesschen et al., 2007). At a global level, estimates of the quantity of N surplus also have great uncertainty. This is evidenced by the more than 50% differences in estimated quantity of N surplus depending on whose estimate was used, as analysed by Zhang et al. (2021).

Each contributing item of the CNB has varying levels of uncertainty with N deposition having the greatest relative uncertainty (CV of ~70%) and crop production having the least uncertainty (CV of ~7%) (Table 3). At the same time, N deposition is a small contributor to the overall N budget with values across the world (as a mean) being less than 10 kg N ha$^{-1}$ year$^{-1}$, so that its contribution to overall uncertainty is also small (Supplementary Material 4). Conversely, items expected to contribute substantially to the overall CNB include synthetic fertilizer use and coefficients for estimating crop nutrient removal. These

items had similar (~20-25%) uncertainty (Supplementary Material 4). The preceding estimates of uncertainty used data that best represented the nutrient component (e.g. CV%'s for maize, rice, soybeans and wheat were used to represent uncertainty in crop nutrient removal since they make up the majority of total grain production worldwide) following IPCC (2006a). It is important to note that there could be greater uncertainties associated with items that were not included in this assessment due to lack of data and/or because it was deemed to make a minor contribution to the overall CNB.

While cropland area has a reasonable 25% estimate of uncertainty, this value does not elucidate the challenges of quantifying the nutrient inputs and outputs from this category of land. Three main issues arise in the current CNB, including 1, it is assumed the same CF values for SF are used to apportion nutrients from manure from livestock to cropland to cropland, 2, no nutrient outputs from herbage removed from some of the categories of cropland (e.g., temporary meadows and pasture or silage maize) are accounted for, and 3, the export of manures between countries is not accounted for. The Netherlands is an example of a

country extremely affected by these limitations of the current methodology. Much of the manure from the dairy sector in the Netherlands is applied on-farm to areas of land growing maize for silage or temporary or permanent meadows and pastures. Yet the proportion of manure applied to cropland may not correspond to the CF values estimated for SF. There is uncertainty in these estimates. In addition, none of the nutrients removed as herbage from the maize for silage or grazed or mown meadows or pastures is included in the total estimate of nutrient outputs. Further, the Netherlands exports 10% of its manure from

livestock to other countries.



Better accounting for N outputs from herbage removed in Dutch 'maize for silage', 'temporary meadows and pastures', or both scenarios combined was estimated to increase NUE from the original 30% in the CNB to 58%, 50% and 77% respectively (Supplementary Material 5). Conversely, accounting for exports of manure from the Netherlands to neighbouring countries was shown to increase NUE from 30 to 32% (Supplementary Material 5). While the Netherlands is an extreme case, other

countries with substantial numbers of livestock and areas of meadows and pastures or fodder crops like maize for silage (e.g. Ireland, Denmark and New Zealand) could also be affected, albeit to a lesser degree (Supplementary Material 5). It must also be noted that the aforementioned scenarios do not account for the confounding effect of manure applied to permanent meadows and pastures, and this could also substantially effect estimates of nutrient surplus and NUE.

**2.3.3 Possible future improvements**

Apart from improving the accuracy and granularity of components that already exist in the CNB, there are several options for future developments of this database.

As highlighted in Section 2.3.2 and Supplemental Material 5, the area of fodder and forage crops in a country can have a substantial effect on nutrient budgets and estimates of nutrient use efficiency. Including estimates of nutrients removed as
fodder and forage crops will therefore allow for a fairer comparison between countries for indicators included in the CNB.

An important future development of the CNB is to account more explicitly for changes in soil nutrient stocks, which are currently 'hidden' in the estimated surpluses or deficits. However, this will be difficult given the dynamic and stochastic characteristics of soil system processes (Cobo et al., 2010).

Including results at a sub-country and crop-specific level is a further area of development. Issues with apportioning fertilizer
and manure to different land use classes or crops will need to be overcome to succeed in spatially disaggregating nutrient budgets and also to accurately estimate separate nutrient budgets for cropland and permanent meadows and pastures. While SF use by crop data are available at a global scale (Ludemann et al., 2022a), estimation of quantities of manure applied to each crop requires suitable survey data that yet do not exist globally. Management of manure during housing and storage before it is applied to cropland also varies spatially and temporally. This can have a substantial effect on the concentrations of nutrients
in manure (Statistics Netherlands, 2012), leading to uncertainty in the quantities of nutrients applied as manure.

The current database does not include estimates of nutrients removed as crop residues, nor are the nutrient concentrations of crop products used in the current database country-specific (see Supplementary Material 3). Progress is being made toward improved predictions of crop harvest index which can be used to determine quantities of crop residues based on quantities of crop products (Ludemann et al., 2022b). However, no studies with global scale are available to indicate what proportion of
crop residues are removed at harvest. This will require extensive collection of survey information to get more relevant crop and country specific values. For improved estimates of nutrients removed as crop products, open databases (www.cropnutrientdata.net; Ludemann et al. (2023a)), and prediction models are being developed to support country and sub-



country specific nutrient concentrations of these crop components. As country and crop specific coefficients (Tier 2 or Tier 3 level) are developed, these can be included in future iterations of the CNB.

Some nutrient inputs currently excluded from the CNB (e.g. nutrients in irrigation water, and nutrients in composted crop residues or human manure) could be included in the future, especially in countries where these constitute a significant contribution to overall inputs.

Finally, with new capabilities becoming available from spatialization of aggregated data to georeferenced grids, the current version and future updates to the database distributed here can provide local-scale information on specific geographic regions,

an information that is generally associated with lower uncertainty specifically in countries with large surface area where only small portions are used for agriculture.

## 3 Results

### 3.1 Global and regional estimates

Global CNB surpluses (i.e. greater nutrient inputs than outputs) were recorded for all three plant nutrients in 2020, with nutrient

loading of N, P, and K progressively increasing over the 1961-2020, period except for K (which decreased by 20% across all cropland since 1961) (Table 4, Figure 2a,b). On a per-hectare basis N, P and K nutrient surpluses changed by 320%, 110% and -27% respectively from 1961 to 2020 (Table 5).

The greatest contributors to nutrient inputs in 2020 were synthetic fertilizers, followed by biological N fixation for N, and manure applied to the soil for P and K (Table 4). The greatest change in any input or output of nutrients (between 1961 and

2020) was the increase in use of SF with changes of 1,000%, 370% and 380% estimated for N, P and K respectively. In 1961 the main nutrient inputs were from livestock manures for all three plant nutrients. With the increase in SF use came a decrease in relative importance of manure as a source of total N inputs. N inputs from manure went from contributing ~38% of total N inputs in 1961 to ~14% in 2020 (Table 4). Over the same period SF went from contributing 22% of total N inputs to 58% (Table 4).

The greatest absolute increases in global N budgets were estimated between 1961 and 1988 (Figure 2a,b), followed by a short-term decrease and then by a less marked increase over the last three decades to 2020. At the same time there was a decreasing trend in N use efficiency from an overall value of 59% in 1961 to 55% in 2020 (Figure 2c). In contrast, the P and K use efficiencies increased over the same period, in particular since the 1980s, from 64% to 75% for P and 46% to 80% for K (Figure 2c). Note, that when nutrient inputs are very low the nutrient use efficiency tends to become higher than 100% which

requires careful interpretation. It may either point at undesirable soil nutrient mining, e.g. in Africa where inputs have been historically low (Vitousek et al., 2009), or it may point at some targeted (desired) soil depletion, e.g. in parts of NW Europe where excessive historical inputs of P led to environmental problems (Einarsson et al., 2020).



The absolute values for nutrient surplus of N (on a total and per hectare basis) were consistently greater than values of P and K surpluses across the 1961-2020 period (Figure 2a,b). K surpluses were consistently greater than P surpluses across the same
period (Figure 2a).

**Table 4: Cropland nutrient budgets for nitrogen (N), phosphorus (P) and potassium (K), by component for years 1961 and 2020 (million tonnes)*.**

| Item | N | | | P | | | K | | |
|---|---|---|---|---|---|---|---|---|---|
| | 1961 | 2020 | % change | 1961 | 2020 | % change | 1961 | 2020 | % change |
| **Inputs** | 45 | 190 | 320 | 8.7 | 27 | 210 | 28 | 58 | 110 |
| Synthetic Fertilizers | 10 | 110 | 1,000 | 4.5 | 21 | 370 | 6.6 | 32 | 380 |
| Manure | 17 | 26 | 53 | 4.3 | 6.6 | 53 | 21 | 25 | 19 |
| Biological Fixation | 11 | 39 | 250 | 0 | 0 | NA | 0 | 0 | NA |
| Atmospheric Deposition | 7.1 | 16 | 130 | 0 | 0 | NA | 0 | 0 | NA |
| **Outputs** | 27 | 100 | 270 | 5.6 | 21 | 280 | 13 | 46 | 250 |
| Crop Removal | 27 | 100 | 270 | 5.6 | 21 | 280 | 13 | 46 | 250 |
| **Soil nutrient surplus** | 18 | **90** | **400** | 3.1 | 6 | **94** | 15 | 12 | -20 |

*Values are rounded to 2 significant figures.

**Table 5: Global cropland nutrient budgets of nitrogen (N), phosphorus (P) and potassium (K), total and by component, for 1961 and 2020 (kg/ha)*.**

| Item | N | | | P | | | K | | |
|---|---|---|---|---|---|---|---|---|---|
| | 1961 | 2020 | % change** | 1961 | 2020 | % change** | 1961 | 2020 | % change** |
| **Inputs** | **33** | **120** | **260** | **6.5** | **18** | **180** | **20** | **37** | **85** |
| Synthetic Fertilizers | 7.7 | 69 | 800 | 3.3 | 13 | 290 | 4.9 | 21 | 330 |
| Manure applied to Soils | 13 | 17 | 31 | 3.2 | 4.2 | 31 | 16 | 16 | 0 |
| Biological Fixation | 7.8 | 25 | 220 | 0 | 0 | NA | 0 | 0 | NA |
| Atmospheric Deposition | 5.3 | 10 | 89 | 0 | 0 | NA | 0 | 0 | NA |
| **Outputs** | **20** | **66** | **230** | **4.1** | **13** | **220** | **9.3** | **29** | **210** |
| Crop Removal | 20 | 66 | 230 | 4.1 | 13 | 220 | 9.3 | 29 | 210 |
| **Soil nutrient surplus** | **13** | **54** | **320** | **2.4** | **5.0** | **110** | **11** | **8** | **-27** |

*Values are rounded to 2 significant figures.

**Percentage difference between the 2020 and the 1961 values over the 1961 value.

Earth System
Science
Data



**Figure 2: The annual cropland nutrient budgets (surpluses if positive or deficits if negative) in millions of tonnes (Mt) of nutrient per FAO Area (plot a), in kilograms of nutrient per hectare (ha) (plot b) and overall nutrient use efficiency percentage (%) (plot c) for different FAO areas of the world for nitrogen (N), elemental phosphorus (P) and elemental potassium (K) from 1961 to 2020.**

## 3.2 Country estimates

There was large heterogeneity in CNB values by country in 2020 (Figure 3). Countries with N, P or K deficits or surpluses greater than the upper (80th) quantile were highlighted red, those with values between the 60th and 80th quantile were highlighted orange, those with values between the 40th and 60th quantile were highlighted yellow, those between the 20th and 40th quantile were highlighted dark green and those below the 20th quantile were highlighted light green. Countries in Africa had cropland



N surpluses less than 40 kg ha$^{-1}$ year$^{-1}$ (with the exception of Egypt with 200 kg N ha$^{-1}$ year$^{-1}$). Most European countries had N surpluses between 40 and 80 kg N ha$^{-1}$ year$^{-1}$, whereas some of the largest values were found in Asia. For instance, China

and India had average N surpluses of 140 kg and 120 kg N  ha$^{-1}$ year$^{-1}$, respectively (Figure 3).  The total number of countries with negative N, P and K surpluses (nutrient deficits) were 14, 64 and 59 in 2020 respectively. It is important to note that extreme values for some countries may represent errors in data collected for those countries (such as for quantities of SF use) rather than or in addition to, actual differences in agronomic performance.

In terms of nutrient use efficiency for 2020, the total number of countries with a nutrient use efficiency greater than 100%

were 14, 64 and 59 respectively for N, P and K (Figure 4). The total number of countries with a nutrient use efficiency less than 50% were 80, 44, and 59 for N, P and K respectively. Combining information from Figure 3 and Figure 4, some countries show differences between their status for the nutrient budget versus nutrient use efficiency. N in Kazakhstan for example is lower ranked in terms of N balance (a deficit of 3 kg N ha$^{-1}$ year$^{-1}$), while it is also ranked highly in terms of N use efficiency (a NUE of 120%), indicating a risk of soil mining. Note, however, that orange colours (efficiencies exceeding 90%) may be

desirable in regions which had large P or K applications historically and are causing environmental problems. Therefore it is important to account for this context when evaluating the NUE of a specific country in the CNB.

Of the 'top 10 countries' ranked based on quantities of synthetic N fertilizer used per country in 2020, four were in Asia (China, India, Pakistan and Indonesia) (Figure 5). Of these top 10 countries, France had the greatest N, P and K surpluses per hectare between 1961-1986 (with a surplus of ~110 kg N ha$^{-1}$ year$^{-1}$ in 1986) (Figure 5a). After this point the China per-hectare N

surpluses became greater than those in France. By 1995 and 2014 China started to have a greater P and K surplus than France, when China had surpluses of 21 kg P ha$^{-1}$ year$^{-1}$ and 35 kg K ha$^{-1}$ year$^{-1}$ (Figure 5a) respectively. There were generally negative trends in N use efficiency for the top 10 countries over the 1961 to 2000 period, after which there was generally a stabilization of annual values (Figure 5b). Exceptions to this negative trend for N were for Brazil and Ukraine, potentially caused by greater harvested areas of N fixing soybeans. There was a greater range in P and K use efficiency over time compared with N use

efficiency (Figure 5b) with, for instance, Indonesia in some years having greater than 200% P use efficiency and greater than 100% K use efficiency between 1961 to 1980.




(a) N

Range

<= 10

10 <= 30

30 <= 60

60 <= 90

> 90

No data

(b) P

Range

<= -3

-3 <= 1

1 <= 5

5 <= 10

> 10

No data

(c) K

Range

<= -7

-7 <= 2

2 <= 20

20 <= 50

> 50

No data

**Figure 3: Cropland nutrient budget deficits (if negative) or surpluses (if positive) (on a kilograms of nutrient per hectare per year basis) for different areas of the world for nitrogen (N) (panel a), elemental phosphorus (P) (panel b) and elemental potassium (K) (panel c) for 2020. Colours are based on quantiles estimated to 2 significant figures. There is considerable uncertainty associated with these data, please refer to Section 2.3.2 for more details. The boundaries and names shown and the designations used on these maps do not imply the expression of any opinion whatsoever on the part of FAO concerning the legal status of any country, territory, city or area or of its authorities, or concerning the delimitation of its frontiers and boundaries. Dashed lines on maps represent approximate border lines for which there may not yet be full agreement. Final boundary between Sudan and South Sudan has not**






**yet been determined. Dotted line represents approximately the Line of Control in Jammu and Kashmir agreed upon by India and Pakistan. The final status of Jammu and Kashmir has not yet been agreed upon by the parties.**

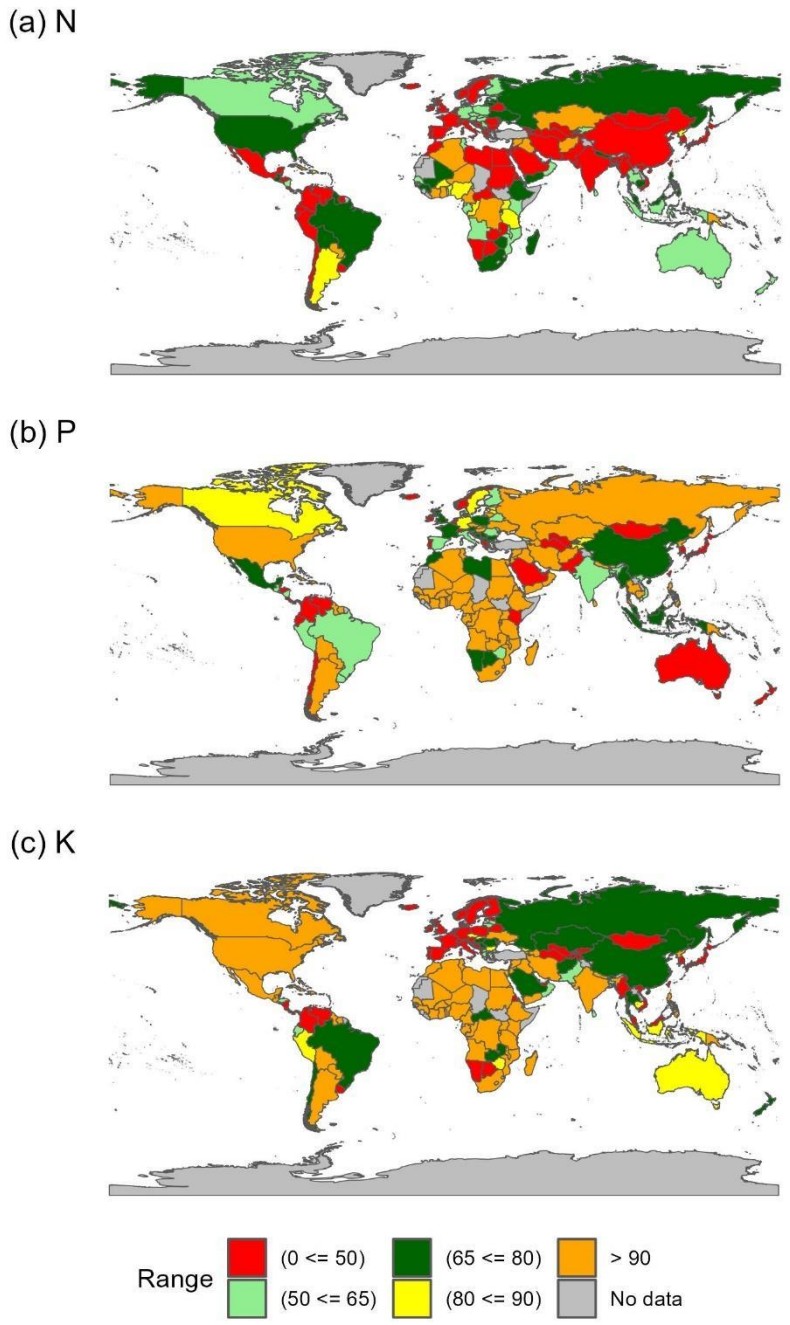

**Figure 4: Cropland nutrient use efficiency (in percent) for different areas of the world for nitrogen (N) (panel a), elemental**
**phosphorus (P) (panel b) and elemental potassium (K) (panel c) for 2020. There is considerable uncertainty associated with these**

data, please refer to Section 2.3.2 for more details. The boundaries and names shown and the designations used on these maps do not imply the expression of any opinion whatsoever on the part of FAO concerning the legal status of any country, territory, city or area or of its authorities, or concerning the delimitation of its frontiers and boundaries. Dashed lines on maps represent approximate border lines for which there may not yet be full agreement. Final boundary between Sudan and South Sudan has not yet been 325 determined. Dotted line represents approximately the Line of Control in Jammu and Kashmir agreed upon by India and Pakistan. The final status of Jammu and Kashmir has not yet been agreed upon by the parties.

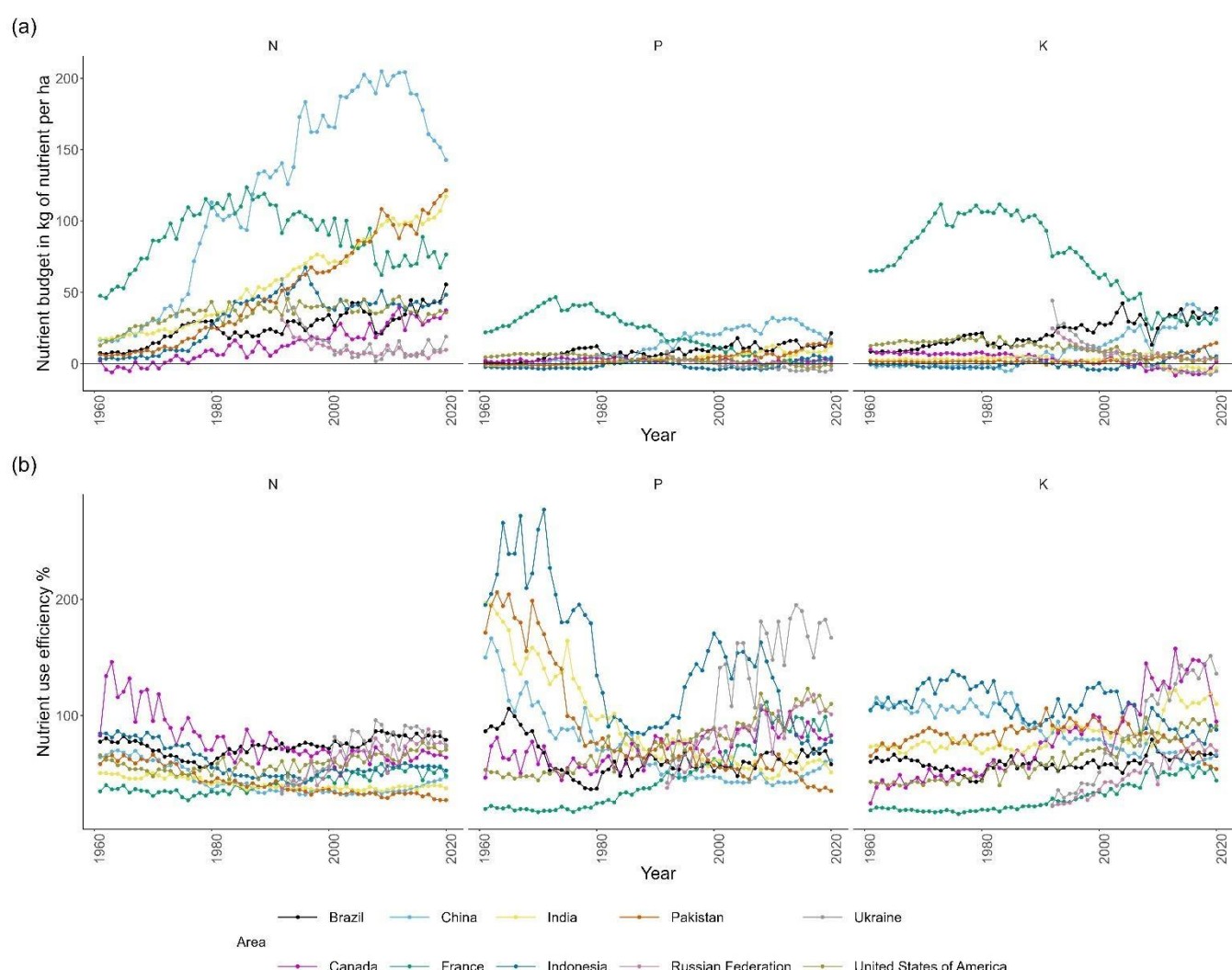

Figure 5: The annual cropland nutrient budgets (surplus if positive and deficit if negative) in kilograms of nutrient per hectare (ha) (panel a) and overall nutrient use efficiency percentage (%) (panel b) for the top 10 countries (based on greatest national nitrogen 330 (N) fertilizer consumption in 2020) for N, elemental phosphorus (P), and elemental potassium (K) for 2020.





## 4. Discussion

Globally, the major trends seen in this analysis include the general increase in nutrient input and outputs of N, P and K during the period 1961-2020, reflecting increased scale and intensity of food production in most countries over the same period. The relative larger increase in growth of inputs vs. outputs has concurrently resulted in greater nutrient surpluses for N and P, while K surpluses decreased. This indicates more emphasis has been placed globally on inputs of N and P compared to K relative to removed nutrients. Many soils still have substantial native K resources and returns on investment for application of K on

cropland are often less than those obtained from applying N and P. Insufficient understanding of how deficient soils are in available K relative to other nutrients may also play a role.

Within the 1961 to 2020 time period the fastest increase in annual N, P and K surpluses occurred between 1961 to 1988. This was followed by a fall in surpluses, followed by relatively stable (for N) or declining trends in (P and K) budgets on a total and per hectare basis. The decline in fertilizer consumption the late 1980s/early 1990s was most likely caused by the breakup

of the former Soviet Union and political changes in much of Eastern Europe (FAO, 2022a). At the same time there was also growing awareness of the environmental effects of unsustainable agricultural management practices in other parts of the world (Cassou, 2018). For example, the European Union (EU) in the late 1980s started implementing policies that reduced direct payments and there was an increase in payments linked to environmental objectives (Cassou, 2018). As a result, the EU N, P and K budgets surpluses decreased over the last three decades. For instance, the EU was estimated to have about 40%, 80%

and 60% decreases in N, P and K budgets on a per hectare basis. These trends initially impacted global trends, though they have been progressively counterbalanced by increasing surpluses in major countries such as China, India, Pakistan and Brazil, largely due to substantial increases in SF use in recent decades. For instance, the application rates in China, India, Pakistan and China increased 230% (as a mean across countries and across N, P and K) between 1990 and 2020, and N, P and K budgets surpluses increased nearly 300% as a mean across those countries over the same period.


### 4.1 Comparisons with previous studies

The general trends in N inputs, outputs, budgets (surpluses/deficits) and use efficiencies over time in the present study were broadly consistent with estimates from previous studies (Figure 6a,b,c and d respectively), with some exceptions. Over the 1961 to 2020 period estimates of N inputs from the current study were 'mid-range' compared with the other studies (Figure

6a), but N outputs were generally greater than those estimated from other studies (Figure 6b). This resulted in estimates of N budgets over time that were mid-range compared with other studies (Figure 6c), and N use efficiencies that were generally greater than estimates from other studies (Figure 6d).



All previous studies estimated the fastest increase in N surpluses between 1961 until around 1988 followed by a drop in N
surpluses for a few years, followed by a less steep increase until 2020 (Figure 6c).

N use efficiency decreased from 1961 until around 1988, followed by an increase in N use efficiency until 2020. Similar trends
over time by the various models in Figure 6 may be attributed to the fact that many of the models used similar sources of data.
For example, 5 of the 10 other models included in Figure 6 used FAOSTAT cropland area data, 8 of the 10 used FAOSTAT
fertilizer use data and at least 4 of the 10 used FAOSTAT crop production data. Because many of the models included in
Figure 6 used similar sources of data, variation in overall N surplus values will not fully account for the variation in and
uncertainty of estimates of key parameters. As described in Section 2.1 some of the most important parameters for estimating
CNB at a country and global level did not have excessively high uncertainty (e.g. cropland area CV% ~25%, crop production
CV%~7% and fertilizer use and crop removal CV%~20%). Parameters with the most uncertainty (e.g. N deposition with a
CV% ~70%) contributed only a small amount to the total N budget (<10 kg N ha$^{-1}$ year$^{-1}$ on average across the world). This
highlights the importance of focussing on refining estimates of the four most influential parameters used in the CNB, namely
cropland areas, crop production quantities, fertilizer use and crop nutrient coefficients.

Estimates of the current study for total N, P and K applied as manure to China were generally less than those estimated using
farmer survey data across the same period by Zhang et al. (2023) (Table 6). Consequently, manure N and P as a percentage of
N and P applied as manure plus synthetic fertilizer from the current study were less than those estimated by Zhang et al. (2023).
Manure K as a percentage of K applied as manure plus SF from the current study was greater than that estimated by Zhang et
al. (2023). The scale of variation in values between the two studies shown in Table 6 are not surprising given the known
uncertainties in estimates of manure and SF application rates for China (Ludemann et al., 2022a). New datasets like those from
Zhang et al. (2023) will be evaluated for how well they may improve the CNB, and where found useful, will be incorporated
into future iterations of the FAOSTAT data product.






**Figure 6: Comparisons of global cropland nitrogen inputs (a), outputs (b), surplus (c) and nitrogen use efficiency (d), 1961-2020, according to various estimates. Non-FAO data (Zhang et al., 2015; Conant et al., 2013; Lassaletta et al., 2014; Mueller et al., 2012; FAO, 2021; Bodirsky et al., 2012; Bouwman et al., 2013; Lassaletta et al., 2016; Lu and Tian, 2017; Nishina et al., 2017) were sourced from Zhang et al. (2021).**






**Table 6: Comparison in mean annual application of manure nitrogen (N), elemental phosphorus (P) and elemental potassium (K) to cropland in China for the period 2005 to 2014 using data from the current study, and Zhang et al. (2023).**

| Data | N manure (million tonnes) | P manure (million tonnes) | K manure (million tonnes) | N in manure as % of N applied as manure plus synthetic fertilizer | P in manure as % of P applied as manure plus synthetic fertilizer | K in manure as % of K applied as manure plus synthetic fertilizer |
|---|---|---|---|---|---|---|
| Current study | 6.9 | 2.1 | 4.7 | 19% | 26% | 31% |
| Zhang et al. (2023) | 4.9 | 1.3 | 4.3 | 14% | 19% | 43% |


## 5 Conclusions

A new reference database on cropland nutrient budgets was detailed in this paper. The data are available in FAOSTAT for the time period 1961 to 2020, with plans for annual updates and continuous methodological improvements. Insights gained from these data include quantification of the hotspot areas from which there may be a surplus or insufficiency in N, P or K nutrients.

For example, all world regions apart from Oceania and Africa showed some, to substantial, N surpluses until 2020. This is a reflection of the broader trend in greater SF N use over that period. However, there were deficits in P and K budgets for Africa and K budgets for the Americas region during the same period. Over time, Europe's relative importance in terms of overall contribution to N budgets were surpassed by Asia (in particular China) in the 1980's. The increasing trends in N surpluses were also shown in other studies, albeit with considerable variation in the absolute values each year, caused by differences in

model set up and sources of data used. Our estimated trends in NUE over time broadly aligned to other studies, except our NUE values were generally greater than those made by other studies. This was a consequence of our estimated N outputs being greater than the other studies. While there was considerable uncertainty (~72% expressed as a CV) associated with some contributing components to the CNB calculation in the present study, in general the components with most uncertainty had least influence on the overall CNB values. The most influential parameters on estimates of CNB included cropland area, crop

production, fertilizer use and crop removal coefficients and should therefore be prioritised for improved accuracy in the future. Further refinements will be an ongoing area of development in future iterations of the FAO CNB.

## 6 Data availability

The CNB data presented from this study covers the period 1961-2020 at the country level, with aggregates made at the regional and global scale. These data are available via

https://datadryad.org/stash/share/Q0cSX1p5HmUR5p2G4RMZQ0DoZmXNNyJ28VSKTFz4Exk (Ludemann et al., 2023b),





and via the FAOSTAT Cropland Nutrient Budget database (https://www.fao.org/faostat/en/#data/ESB). R code used to create

tables and figures in this article can be accessed via the following git repository: https://github.com/ludemannc/fao_cnb.

Further information on the derivation of cropland fraction estimates for N, including our analytical code and accompanying

technical note, can be accessed via the following git repository: https://github.com/KEJackson-94/Fr_Crop_Estimates.

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

555

**Author contributions**

CL collated manure and crop nutrient removal coefficients, performed analysis of data and wrote draft article. NW, GO and FNT developed UN FAO cropland budget on the https://www.fao.org/faostat/en/#data/ESB website, performed analysis of data, and wrote draft article. SV analysed N deposition data and wrote N deposition section. RE analysed the biological N 560 fixation data and wrote the biological N fixation section. AG analysed fertilizer use data. KJ and XZ analysed fraction of N fertilizer applied to croplands data.

All authors were part of a UN FAO cropland nutrient budget steering group who determined how the database was developed. They also all edited and approved the final article for submission.



**Competing interests**

565     One author (FNT) is a member of the editorial board of *Earth Systems Science Data*. AD and CL received financial support

from the International Fertilizer Association. FNT and NW acknowledge funding from the Swiss Federal Office of Agriculture,

which made this work possible.