# Peer review of "A global reference database in FAOSTAT of cropland nutrient budgets and nutrient use efficiency: nitrogen, phosphorus and potassium, 1961-2020."

_Earth System Science Data, 2023_

## Referee Comment (RC2)

Comments to the Author

This manuscript focuses on nutrient budgets and nutrient use efficiency, present a global database of country-level budget estimates for nitrogen (N), phosphorus (P) and potassium (K) in cropland. This study introduces improvements over previous work in relation to key nutrient coefficients affecting nutrient budgets and use efficiency. Results highlight the wide range in nutrient use and use efficiencies across geographic regions, nutrients, and time. However, before acceptance, several issues deserve attention, as outlined below.

General:

(1) The necessity and innovation of the article should be presented to the introduction.

(2) This study is not explicitly addressed in its exploration of the global farmland nutrient budget versus nutrient use efficiency, as many studies have been conducted in this area;

(3) In the process of calculating cropland nutrient budgets and nutrient use efficiency, many coefficients are used consistently, which may lead to great uncertainty;

(4) For the input of organic nitrogen, the CF value of organic fertilizers is the same as that of synthetic fertilizers, which may cause certain deviations in the results;

(5) Whether to consider adding a part, compared with other research methods, the necessity and innovation of this study.

(6) The discussion section lacks sufficient elaboration on key findings and the content appears too vague.

(7) Show more self-criticism towards your methods, discuss all limitations of your findings.

Specific:

(8) Line 50-55 "We see two main rationales for estimating nutrient budgets on cropland. First, cropland is typically where nutrient flows and related environmental impacts are highest, and cropland budgets and derived indicators such as the surplus are therefore more likely to capture potential pollution hotspots. Second, permanent meadows and pastures present some particular method challenges, primarily due to lack of global data on productivity and biological N fixation" Please add relevant references.

(9) Line 100-115 "For the majority of countries, due to lack of specific information, default cropland fraction estimates of 100% were used for N, P, and K, thereby assuming all fertilizers were applied on cropland area." Is there evidence to support this hypothesis?

(10) Line 115-120 It is assumed the same CF values for SF are used to apportion nutrients from manure from livestock to cropland to cropland. For example, the proportion of animal manure returning to the field like horses seems to be very low, and the rationality of this parameter is doubted.

(11) Line175-180 Fraction of livestock manure applied to cropland, The uncertain of livestock manure should also be considered.

(12) Line360-365 N inputs from the current study were 'mid-range' compared with the other studies but N outputs were generally greater than those estimated from other studies, This result requires careful interpretation.

(13) Line 360-375 Compared with previous studies, only nitrogen analysis, lack of phosphorus, potassium analysis

(14) Line405-410 NUE values were generally greater than those made by other studies, this result requires careful interpretation.

---

## Author Comment (AC1)

**Authors reply to comments on essd-2023-206 made by two reviewers Peiyu Cao (Reviewer 1, https://doi.org/10.5194/essd-2023-206-RC1) and an Anonymous Reviewer (Reviewer 2, https://doi.org/10.5194/essd-2023-206-RC2).**

**We wish to thank the reviewers for taking the time to provide feedback on our manuscript and for their constructive comments. Comments from each reviewer are included below followed (in bold) by our changes made to the manuscript based on their feedback.**

**Reviewer 1:**

Review of "A global reference database in FAOSTAT of cropland nutrient budgets and nutrient use efficiency: nitrogen, phosphorus and potassium, 1961-2020" by Ludemann et al.

Essd-2023-206

I enjoyed the paper and I am confident it will be highly influential in both the research and management communities. I have some suggestions for improving the readability of the paper, enhancing the description of the methodology, and refining interpretation of the results . I recommend acceptance pending minor revision, as detailed below.

**Thanks for your positive feedback on our manuscript!**

Major comments:

I think the readability of the paper can be improved by maintaining consistent terminology regarding nutrient balance, nutrient surplus, and nutrient deficit. It would be helpful to provide a clear definition at the beginning of the paper. For instance, nutrient surplus in line 38 denotes nutrient balance, which can be either positive or negative. In lines 58-59, nutrient surplus is directly defined as balance, and deficit is defined as negative surplus. However, in the following content, the surplus and deficit are explained as positive and negative values respectively. I would suggest to give a definitive clarification in Abstract (line 22) and 38, such as: "The nutrient balance (surplus if positive or deficit if negative)" and eliminate the verbose and inconsistent explanations for surplus and deficit elsewhere.

**We have now given a definitive clarification in the Abstract (line 22):**

- **"nutrient balance (surplus if positive or deficit if negative)"**

**and in the Introduction section on line 38 as follows:**

- **"The nutrient balance (defined as the difference between nutrient inputs and productive outputs; termed a surplus if positive and a deficit if negative),"**

The paragraph between line 56 and 60 explains the components of nutrient inputs and outputs, and calculation of nutrient balance and nutrient use efficiency in this study. It would be better to merge this paragraph to Method section, especially for the calculation of nutrient use efficiency, which is lacking in Method.

**We now include this description in the Method section as follows on line 75:**

- **"The nutrient budget inputs in cropland considered in this work included the application of synthetic fertilizers (SF) (also referred to as "chemical fertilizers" or "mineral fertilizers"), manure from livestock, the N inputs through biological N**

**fixation, and the atmospheric N deposition. The nutrient budget outputs were the nutrients removed via crop harvest. The nutrient budget balance was calculated as the difference between inputs and outputs. Nutrient use efficiency was computed as nutrient outputs as a percentage of nutrient inputs."**

In the paper, it is mentioned that N amountsin manure are estimated by multiplying live animal weights by N excretion coefficients (line 119-121), which represents the amount of N livestock produces (N excretion). However, according to the methodological note of "Manure applied to soils" domain under "Climate Change-Emissions-Farm gate" by FAOSTAT, it is the N in treated manure in manure management systems that is applied to soils. I assume the manure N applied to soil in this study is consistent to that from FAOSTAT, I would suggest to revise the description regarding manure N to avoid confusion and ensure consistency.

**Thanks for picking up on this error in the text we have changed this to as follows on line 126:**

- **"Organic N inputs were limited to manure applied to cropland soils (MAS). MAS was estimated as N from treated manure in manure management systems applied to soil following IPCC Guidelines for National Greenhouse Gas Emission Inventories at Tier 1 (e.g., FAO (2022e))."**

The spatial patterns of nutrient budget and nutrient use efficiency are very interesting. As a reader, I expect to compare the spatial pattern of single nutrient input and output with the nutrient budget and nutrient use efficiency. I would suggest to incorporate these maps in supplementary.

**We have now added Supplementary Material 6 with these maps and refer to this Supplementary material on line 297 as follows:**

**"Maps of the total N, P and K inputs and outputs are available in Supplementary Material 6".**

The authors point out that the N outputs are generally greater than those from other studies, resulting in a greater NUE in line 359-362.To help readers interpret the findings appropriately and assess potential uncertainties associated with using this data, I would suggest to explain the factors contributing to this difference.

**We now better explain the factors contributing to this difference on line 380:**

- **"Multiple factors could have contributed to the inter-study variation in indicators shown in Figure 6. Firstly, FAOSTAT crop production and fertilizer data have been updated since the previous studies were published. Any changes in historic crop production and fertilizer input data will contribute to differences in estimates of total N outputs and N inputs respectively. To put this into context, Zhang et al., (2021) indicated the FAOSTAT data for China's N fertilizer use was 10 million tonnes per year lower based on the 2017 version of the data compared with the 2000 version. In addition, variation in estimates of the N concentration of crop product for each crop species between studies will result in variation in estimated N outputs. A summary of existing parameters of N content by crop type has shown large divergence among studies (Zhang et al., 2020), and some studies also do not account for the N content in the crop types that have limited data. Taking advantage of existing data, the present study developed and used gap-filled crop product nutrient concentrations, while future research is needed to improve the availability and quality of such data.**

Figure 6 from Zhang et al. (2020) showing variation nutrient concentrations of crop products.

[Figure]

**Figure 6.** N (a) and P (b) contents of 11 crop or crop groups. N contents are from five data sources (Bodirsky et al., 2012; Bouwman et al., 2017; Feedipedia, 2012; IPNI, 2014; Lassaletta, Billen, Grizzetti, Garnier, et al., 2014), and P contents are from seven data sources (AUSNUT, 2013; Bouwman et al., 2017; FAO, 2006; Feedipedia, 2012; Gourley et al., 2010; IPNI, 2014; USDA, 2013).

Minor comments:

The paper lacks reference information for Buckley et al. 2018 in Figure 1.

**The Buckley reference has now been added to the reference list and the figure title along with all the other references used in Figure 1 to make it easier for the reader to track references.**

Many of the reference are difficult to find, such as the reference of FAO. Please update valid reference for readers to access

**The FAOSTAT and IFASTAT references from Figure 1 have been made more explicit in the figure and the title of the figure and are all included in the reference list. We have now checked all the references to make sure they have valid reference information.**

Line 187: typo "to cropland to cropland"

**We corrected this typo as per line 196**.

Line 370: incomplete sentence.

**Corrected this incomplete sentence on line 395:**

- **"Many of the models included in Figure 6 used similar sources of data, therefore variation in overall N balance values will not fully account for the variation in and uncertainty of estimates of key parameters."**

Table 6: The estimates given in the first row were by Zhang et al. (2023) but not from current study.

**We double checked this and we made a change as follows on Table 6:**

- **"The top row of data is now correctly labelled Zhang et al (2023)"**

Citation: https://doi.org/10.5194/essd-2023-206-RC1

**Reviewer 2**

Comments to the Author

This manuscript focuses on nutrient budgets and nutrient use efficiency, present a global database of country-level budget estimates for nitrogen (N), phosphorus (P) and potassium (K) in cropland. This study introduces improvements over previous work in relation to key nutrient coefficients affecting nutrient budgets and use efficiency. Results highlight the wide range in nutrient use and use efficiencies across geographic regions, nutrients, and time. However, before acceptance, several issues deserve attention, as outlined below.

General:

(1) The necessity and innovation of the article should be presented to the introduction.

**We have now elaborated in the Introduction more on the necessity and innovation of this article/dataset as per line 46:**

- **"Time series data showing temporal changes are essential to monitoring progress toward nutrient related goals (Zhang et al., 2021). Some nutrient budget time series with global scope have been published. However, to the authors knowledge they have been heavily biased to N (Zhang et al., 2015; Conant et al., 2013; Lassaletta et al., 2014; Mueller et al., 2012; Bouwman et al., 2017; FAO, 2021; Bodirsky et al., 2012; Bouwman et al., 2013; Lassaletta et al., 2016; Lu and Tian, 2017; Nishina et al., 2017), few have been published for P and no time series for K has been published meaning no studies or datasets have integrated all three nutrients into a long-term nutrient budget database".**

(2) This study is not explicitly addressed in its exploration of the global farmland nutrient budget versus nutrient use efficiency, as many studies have been conducted in this area;

**We thank the reviewer for raising this point. We consider this comment relates to point 1 from Reviewer 2 where the question of novelty (innovation) of the data article was raised. As per our reply to Reviewer 2, point 1 we have made the innovation/novelty to our work more clear in the Introduction.**

(3) In the process of calculating cropland nutrient budgets and nutrient use efficiency, many coefficients are used consistently, which may lead to great uncertainty;

**We have checked our coefficients for inconsistency and they appear to be consistent. However we agree with Reviewer 2's point 4 (below) that use of the same CF value for manure and synthetic fertilizers is a limitation of our study and is an area of uncertainty. This is why we devoted section 2.3.3 and Supplementary Material 5 to uncertainties. Supplementary material 5 in particular provides scenario analysis and discussion of how CF can cause deviations of the results for countries that are most affected by this coefficient.**

(4) For the input of organic nitrogen, the CF value of organic fertilizers is the same as that of synthetic fertilizers, which may cause certain deviations in the results;

**This is a limitation of our work. In future iterations of this database we aim to improve these coefficients for manure. We therefore devote a paragraph in section 2.3.2 (Uncertainty), line 195 to CF:**

- **"Three main issues arise in the current CNB, including 1, it is assumed the same CF values for SF are used to apportion nutrients from manure from livestock to cropland, 2, no nutrient outputs from herbage removed from some of the categories of cropland (e.g., temporary meadows and pasture or silage maize) are accounted for, and 3, the exchange of manure between countries is not accounted for. The Netherlands is an example of a country extremely affected by these limitations of the current methodology. Much of the manure from the dairy sector in the Netherlands is applied on-farm to areas of land growing maize for silage or temporary or permanent meadows and pastures. Yet the proportion of manure applied to cropland may not correspond to the CF values estimated for SF. There is uncertainty in these estimates.**
- **We also devote Supplementary Material 5 to describing the effects of changing some of these assumptions for key countries.**

(5) Whether to consider adding a part, compared with other research methods, the necessity and innovation of this study.

**As per our reply to Reviewer 2's comment 1, we have now elaborated in the Introduction on the necessity and novelty of this article/dataset.**

(6) The discussion section lacks sufficient elaboration on key findings and the content appears too vague.

**Given the world scope of this data article we could not get into too much detail of the results for each country, so had to elaborate at the region level. The exception for this was for China where we did go into more detail given the importance of this country to global nutrient budgets (see Table 6 and text in lines above this). We also detail the limitations of our method for a selected group of countries most affected by those limitations (see Supplementary Material 5) and from line 198 as follows:**

- **"The Netherlands is an example of a country extremely affected by these limitations of the current methodology. Much of the manure from the dairy sector in the Netherlands is applied on-farm to areas of land growing maize for silage or temporary or permanent meadows and pastures. Yet the proportion of manure applied to cropland may not correspond to the CF values estimated for SF. There is uncertainty in these estimates. In addition, none of the nutrients removed as herbage from the maize for silage or grazed or mown temporary meadows or pastures is included in the total estimate of nutrient outputs. Further, the Netherlands exports 10% of its manure from livestock to other countries. "**

(7) Show more self-criticism towards your methods, discuss all limitations of your findings.

**We thank the reviewer for this comment. In section 2.3.3 (Possible future improvements) we now more explicitly discuss the limitation of using the same values for fraction of livestock manure that is applied to cropland and fraction of total agricultural fertilizer applied to cropland. On line 229 we include:**

**"In addition, use of the same value for fraction of N fertilizer applied to cropland as that used for fraction of livestock manure applied to cropland introduces uncertainty to the overall CNB estimates. As described in Supplementary Material 5, this assumption may not hold for every country. Introduction of country-specific fractions representing the proportion of manure from livestock that is applied to cropland will be an important improvement in future iterations of the CNB."**

**In terms of other limitations/uncertainties/self-criticism of our methods we have chosen to include these in section 2.3 (Data limitations and uncertainty). In this section we provide a detailed discussion of the scope, uncertainties and possible future improvements for our method/database.  In Supplementary Material 4 we quantify the uncertainties of components that contribute to our overall method. In Supplementary Material 5 we discuss in detail some of the limitations of our method for a selection of case-study countries most impacted by nuances of our method (mainly to do with livestock manure and areas of temporary meadows and pastures and maize silage).**

**We have also added limitations to the conclusion as per line 463:**

**"It is also important to note that for some countries limitations of availability of data could have a substantial effect on estimates of overall nutrient budget or nutrient use efficiency for cropland. This is especially important in relation to how nutrients are assigned to areas of forage and fodder crops as well as exports of manure from livestock to other countries and manure application to permanent meadows and pastures."**

Specific:

Line 50-55 "We see two main rationales for estimating nutrient budgets on cropland. First, cropland is typically where nutrient flows and related environmental impacts are highest, and cropland budgets and derived indicators such as the surplus are therefore more likely to capture potential pollution hotspots. Second, permanent meadows and pastures present some particular method challenges, primarily due to lack of global data on productivity and biological N fixation" Please add relevant references.

**We have now added references to the paragraph starting on line 58 as follows:**

**"We see two main rationales for estimating nutrient budgets on cropland. First, cropland is typically where nutrient flows and related environmental impacts are the highest, and cropland budgets and derived indicators such as the surplus are therefore more likely to capture potential pollution hotspots (West et al., 2014). Second, permanent meadows and pastures present some particular method challenges, primarily due to lack of global data on productivity and biological N fixation (Tubiello et al., 2023; Schils et al., 2013). "**

(9) Line 100-115 "For the majority of countries, due to lack of specific information, default cropland fraction estimates of 100% were used for N, P, and K, thereby assuming all fertilizers were applied on cropland area." Is there evidence to support this hypothesis?

**Yes. In the paragraph that followed the mentioned sentence (line 113) we went through data from all countries to see if there was reasonable evidence to convince us there was a cropland fraction (CF) that was significantly less than 100%. Only countries where we had reasonable evidence to suggest a CF less than 100% were given such a (smaller) number. Evidence was based on a mixture of fertilizer use by crop data as well as data given directly by two countries (New Zealand and Ireland). Detailed methodology for estimating the CF values is included in Supplementary Material 1.**

(10) Line 115-120 It is assumed the same CF values for SF are used to apportion nutrients from manure from livestock to cropland to cropland. For example, the proportion of animal manure returning to the field like horses seems to be very low, and the rationality of this parameter is doubted.

**Future iterations of the FAO cropland nutrient budget will aim to improve the way we apportion nutrients from manure from livestock to cropland. Unfortunately, at this stage we have insufficient data to better estimate the fraction of manure applied to cropland by country than that based on the CF for synthetic fertilizer. See also changes we made to comment (11) below.**

(11) Line175-180 Fraction of livestock manure applied to cropland, The uncertain of livestock manure should also be considered.

**Lines 176 onwards and Supplementary Material 5 describe the uncertainty of the fraction of livestock manure applied to cropland and highlight that for some countries the fraction of fertilizer applied to cropland may not be a good indicator of fraction of manure from livestock applied to cropland.**

**However, we added the following to the 'Possible future improvements' section on line 229:**

**"In addition, use of the same value for fraction of N fertilizer applied to cropland as that used for fraction of livestock manure applied to cropland introduces uncertainty to the overall CNB estimates. As described in Supplementary Material 5, this assumption may not hold for every country. Introduction of country-specific fractions representing the proportion of manure from livestock that is applied to cropland will be an important improvement in future iterations of the CNB."**

(12) Line360-365 N inputs from the current study were 'mid-range' compared with the other studies but N outputs were generally greater than those estimated from other studies, This result requires careful interpretation.

**We have now added further interpretation from line 380:**

**"Multiple factors could have contributed to the inter-study variation in indicators shown in Figure 6. Firstly, FAOSTAT crop production and fertilizer data have been updated since the previous studies were published. Any changes in historic crop production and fertilizer input data will contribute to differences in estimates of total N outputs and N inputs respectively. To put this into context, Zhang et al., (2021) indicated the FAOSTAT data for China's N fertilizer use was 10 million tonnes per year lower based on the 2017 version of the data compared with the 2000 version. In addition, variation in estimates of the N concentration of crop product for each crop species between studies will result in variation in estimated N outputs. A summary of existing parameters of N content by crop type has shown large divergence among studies (Zhang et al., 2020), and some studies also do not account for the N content in the crop types that have limited data. Taking advantage of existing data, the present study developed and used gap-filled crop product nutrient concentrations, while future research is needed to improve the availability and quality of such data."**

(13) Line 360-375 Compared with previous studies, only nitrogen analysis, lack of phosphorus, potassium analysis

**Unfortunately we could not find equivalent long term timeseries data for potassium for comparisons to be made (this highlights the novelty of our dataset). However, for P budgets we found time series data from Zou et al. (2022). We therefore added Figure 7 and text in lines 423 to compare results:**

**"The general trends in P inputs, outputs, balances (surpluses/deficits) and use efficiencies over time in the present study were broadly consistent with estimates from Zou et al. (2022) (Figure 7a,b,c and d respectively). However, P inputs and outputs and PUE estimated in the current study were generally greater than those estimated by Zou et al. (2022). Concurrently the P surplus was estimated as being less in the current study than Zou et al. (2022) and the difference in estimates increased after 1990 and especially after 2008 when the Zou et al. (2022) estimates became substantially greater than our current estimates.**

**Zou et al. (2022) used the same FAO (2022d) areas of cropland and fertilizer input values as was used in the current study, indicating crop P removal is the main contributor to these differences in values. Estimates of the concentration of P in crop products used in the present study were generally greater than those used by Zou et al. (2022). This explains why crop P removal (outputs) and PUE in the present study are greater than those estimated by Zou et al. (2022). For example, of the major crops in the current study, rice, soybeans and maize had 12%, 30% and 18% greater P concentrations than Zou et al. (2022). Concentrations of P in wheat and barley in the current study were estimated as being 4% and 2% less than that used by Zou et al. (2022).**

**A reason why estimates of P inputs by Zou et al. (2022) are less than the current study is that Zou et al. (2022) used a different method for assigning the fraction of total fertilizer used in agriculture to cropland. Zou et al. (2022) assumed that the fractions of P fertilizer used for cropland are the same as fractions of N fertilizer used for cropland following Zhang et al. (2015). In addition, the FAO updated its fertilizer input data since the Zou et al. (2022) study was published. This may have also contributed to these differences in P inputs."**

(14) Line405-410 NUE values were generally greater than those made by other studies, this result requires careful interpretation.

**Please see text added based on feedback from Reviewer 2 comment 12 where we elaborate on why there may be differences.**

**References**

Zhang, X., Davidson, E. A., Zou, T., Lassaletta, L., Quan, Z., Li, T., & Zhang, W. (2020). Quantifying Nutrient Budgets for Sustainable Nutrient Management. *Global Biogeochemical Cycles*, *34*(3), e2018GB006060. https://doi.org/10.1029/2018gb006060

Zou, T., Zhang, X., & Davidson, E. A. (2022). Global trends of cropland phosphorus use and sustainability challenges. *Nature*, *611*(7934), 81-87. https://doi.org/10.1038/s41586-022-05220-z

---

## Author Response (AR2)

**Authors reply to comments on essd-2023-206 made by Topic Editor Zhen Yu on the 17th October 2023.**

**We wish to thank the editor (Zhen Yu) for taking the time to provide feedback on our manuscript and for his constructive comments. Comments by Zhen are included below followed (in bold) by our changes made to the manuscript based on his feedback.**

17 Oct 2023

**Topic editor decision: Reconsider after major revisions**

by Zhen Yu

I have a few suggestions:
1. Please note that your data set DOI and its in-text citation must be provided in the abstract.

**As per line 37 of the abstract we have now added:**

**"Information from the present study is available from DOI https://datadryad.org/stash/dataset/doi:10.5061/dryad.hx3ffbgkh (Ludemann et al., 2023b) as well as at the FAOSTAT database (https://www.fao.org/faostat/en/#data/ESB) (FAO, 2022a), with annual updates.**

2. For consistency in Figure 1, please use "AD" instead of "ND." I would also suggest creating a more visually appealing conceptual diagram for Figure 1.

**For brevity and considering the information in Figure 1 is available in Equation 1 and as text in the Methods section we decided to delete this figure and any referral to it.**

3. When deriving nutrient balances per unit of area, have you included fallow land in the calculation?

**We used the area of cropland as the denominator for deriving nutrient balances per unit area. As per line 93 cropland is defined as:**

**"'land used for cultivation of temporary and permanent crops in rotation with fallow, meadows and pastures within cycles of up to five years.'".**

4. I recommend restructuring your Results and Discussion sections. Why is there only a "4.1 Comparisons with previous studies" section in your Discussion? The figures and tables comparing your results with previous studies may be more suitable in the Results section.

**We have now created a combined Results and Discussion section where we include:**

**3. Results and Discussion**
**3.1 Global and regional estimates (same as previous version of manuscript)**
**3.2 Country estimates (same as previous version of manuscript)**
**3.3 Major trends (I used text from lines 353 to 372 from the previous version of the discussion section)**
**3.4 Comparisons with previous studies (used text and figures/tables from the section named '4.1 Comparisons with previous studies' from the previous version of the manuscript)**
**4.0 Conclusions (same as current manuscript)**